# Consolidated BRCA1/2 Variant Interpretation by MH BRCA Correlates with Predicted PARP Inhibitor Efficacy Association by MH Guide

**DOI:** 10.3390/ijms21113895

**Published:** 2020-05-29

**Authors:** Yosuke Hirotsu, Udo Schmidt-Edelkraut, Hiroshi Nakagomi, Ikuko Sakamoto, Markus Hartenfeller, Ram Narang, Theodoros G. Soldatos, Sajo Kaduthanam, Xiaoyue Wang, Stephan Hettich, Stephan Brock, David B. Jackson, Masao Omata

**Affiliations:** 1Genome Analysis Center, Yamanashi Central Hospital, Kofu 400-8506, Japan; m-omata0901@ych.pref.yamanashi.jp or; 2Molecular Health GmbH, Research Department, 69115 Heidelberg, Germany; udo.schmidt-edelkraut@molecularhealth.com (U.S.-E.); markus.hartenfeller@molecularhealth.com (M.H.); ramnarang@gmx.de (R.N.); theodoros.soldatos@molecularhealth.com (T.G.S.); sajo.kaduthanam@gmx.de (S.K.); xiaoyue.wang@molecularhealth.com (X.W.); stephan.hettich@molecularhealth.com (S.H.); stephan.brock@molecularhealth.com (S.B.); david.jackson@molecularhealth.com (D.B.J.); 3Department of Breast Surgery, Yamanashi Central Hospital, Kofu 400-8506, Japan; h-nakagomi@ych.pref.yamanashi.jp; 4Department of Obstetrics and Gynecology, Yamanashi Central Hospital, Kofu 400-8506, Japan; i-nagata@ych.pref.yamanashi.jp; 5University of Tokyo, 7-3-1 Hongo, Bunkyo-ku, Tokyo 113-8655, Japan

**Keywords:** *BRCA1*, *BRCA2*, PARP inhibition, HBOC, cancer

## Abstract

*BRCA1/2* variants are prognostic biomarkers for hereditary breast and/or ovarian cancer (HBOC) syndrome and predictive biomarkers for PARP inhibition. In this study, we benchmarked the classification of *BRCA1/2* variants from patients with HBOC-related cancer using MH BRCA, a novel computational technology that combines the ACMG guidelines with expert-curated variant annotations. Evaluation of *BRCA1/2* variants (*n* = 1040) taken from four HBOC studies showed strong concordance within the pathogenic (98.1%) subset. Comparison of MH BRCA’s ACMG classification to ClinVar submitter content from ENIGMA, the international consortium of investigators on the clinical significance of *BRCA1/2* variants, the ARUP laboratories, a clinical testing lab of the University of UTAH, and the German Cancer Consortium showed 99.98% concordance (4975 out of 4976 variants) in the pathogenic subset. In our patient cohort, refinement of patients with variants of unknown significance reduced the uncertainty of cancer-predisposing syndromes by 64.7% and identified three cases with potential family risk to HBOC due to a likely pathogenic variant *BRCA1* p.V1653L (NM_007294.3:c.4957G > T; rs80357261). To assess whether classification results predict PARP inhibitor efficacy, contextualization with functional impact information on DNA repair activity were performed, using MH Guide. We found a strong correlation between treatment efficacy association and MH BRCA classifications. Importantly, low efficacy to PARP inhibition was predicted in 3.95% of pathogenic variants from four examined HBOC studies and our patient cohort, indicating the clinical relevance of the consolidated variant interpretation.

## 1. Introduction

Deleterious *BRCA1/2* variants have emerged as critical prognostic and predictive biomarkers for deficiency in DNA damage repair by the homologous recombination (HR) machinery [1], particularly for patients with hereditary breast and/or ovarian cancer syndrome (HBOC) but also for sporadic cases [2,3,4]. Over the past decade, the poly (ADP-ribose) polymerase (PARP) family of proteins has been established as an important therapeutic target [5,6,7,8]. PARPs are involved in DNA repair of single-strand DNA breaks and inhibition of PARP function is based on the concept of synthetic lethality in cancers with simultaneous HR deficiency [9]. Despite the utility of platinum sensitivity and *BRCA1/2* variants as predictors of treatment response, there is an urgent need for greater resolution in the predictive capacity of individual *BRCA1/2* variants to understand the determinants of response related to the degree of HR deficiency. An additional caveat in understanding the clinical significance of detected variants is the enormous diversity of variants that can exist in the general population. To aid this process, the American College of Medical Genetics and Genomics (ACMG) has provided a set of guidelines that enable the classification of variants [10]. Expanding upon this framework, Molecular Health (MH) developed MH BRCA, a technology that supports the clinical interpretation of *BRCA1/2* variants by combining the rules laid out in the ACMG guidelines with a proprietary database of expert curated variant annotations. Importantly, MH BRCA also integrates variant population-frequency (PF) data from the Japanese genomic cohort of the Tohoku Medical Megabank Organization (ToMMo) [11].

In this study, we focused retrospectively on 346 patients (334 with breast and/or ovarian cancer and 12 with prostate cancer) sequenced at the Yamanashi Central Hospital (YCH) with the purpose to compare our *BRCA1/2* variant interpretation provided by the current clinical workflow with those provided through reanalysis with the MH BRCA technology and to evaluate the predicted response to PARP inhibition with the treatment decision support service MH Guide. The study design is schematically depicted in Figure 1.

## 2. Results

### 2.1. Pre-Evaluation of MH BRCA

We first tested the diagnostic support of the tool by comparing *BRCA1/2* classifications from a case/control study on 7051 Japanese HBOC patients and 11,241 controls [13]. Momozawa et al. classified variants of 11 breast cancer genes based on the ruleset of the ACMG guideline [10]. This study used the 3-Tier system for the classification of pathogenic, benign, and variants of unknown significance (VUS). A total of 838 unique *BRCA1/2* variants were pooled from the female and male dataset. As MH BRCA provides classifications from reputable variant annotation databases (DB), we examined the public accessible knowledge in this dataset. The DB consensus classification, summarizing variant annotations from the expert consortia ENIGMA and BIC, both from BRCA exchange [16], ClinVar [17], and the ARUP Laboratories of the University of Utah (http://arup.utah.edu/database/BRCA/), demonstrated that 40.5% of variants were not documented in the public domain (Figure 2A and Appendix A). In contrast, ACMG-guided classification was possible on the full dataset and revealed in the cluster analysis of the pathogenic subset a strong concordance of 96.8% with Momozawa et al. (Figure 2B). Analysis of the benign subset showed an unsatisfying low performance in the ACMG-guided interpretation, whereas combined with the DB consensus classification, it increased the accuracy in prediction of benignity to 95.3% concordance, underlining the benefit of the consolidated variant interpretation (Figure 2C). In the largest subset of undetermined VUS, we found in 41.9% of cases reclassification by MH BRCA (Figure 2D). In summary, the consolidated variant interpretation demonstrated strong concordance with clinical assessed *BRCA1/2* classifications and indicated the potential for enhanced VUS reclassification.

### 2.2. Impact of Japanese Genomic Cohort Data

We next investigated the contribution of Japanese-specific PF data from ToMMo for the determination of benign variants in Japan. As Momozawa et al. used the ToMMo data for assessment of benign polymorphisms, we selected three additional Japanese HBOC studies. To this end, we used the study of Nakagomi et al. [18] with clinical assessed interpretations of *BRCA1/2* variants and studies of Hirotsu et al. [15] and Arai et al. [14] with classifications from FALCO Biosystems using the license of Myriad Genetics (Salt Lake City, UT, USA). Benign classifications were either based on the clinical occurrence-rate or reported test results, whereas Hirotsu et al. used in addition the Japanese-specific HGVD database of Kyoto University, Japan [12]. *BRCA1/2* variants reported with an allele frequency >1% in an outbred population cohort are considered as benign in accordance with the ENIGMA classification criteria. A comparison of each study with MH BRCA in the absence or presence of ToMMo data with PF > 1% revealed an improvement exclusively in the benign category (Figure 2E and Appendix A). Cluster analysis in the pooled dataset was associated with consistent pathogenic or likely pathogenic interpretation (Figure 2F). Analysis of the pooled benign/likely benign subset demonstrated that adding Japanese-specific PF data to MH BRCA increased the benign classifications by 10.5%, highlighting the improvement due to integrated ToMMo genomic cohort data (Figure 2G). Reclassification of VUS was found in 22 out of 48 variants (Figure 2H and Appendix A).

To assess the importance of Japanese-specific PF data, we compared ToMMo with the 1000 Genomes Project data (1000G) [19], the Exome Aggregation Consortium (*ExAC*) [20], and the Genome Aggregation Database (GnomAD) [21]. The Japanese cohort-size of ToMMo includes 3554 individuals and is 30-fold higher than from the 1000G data (120 of 4888 individuals), whereas ExAC and GnomAD data used a general East-Asian population cohort. Comparison based on the pooled benign subset from the examined four Japanese HBOC studies showed that assessment of benign variants was in 33 out of 176 variants (19%) based on reliable data with PF > 1% (Appendix A). Although ToMMo and 1000G had similar performance in the Japanese-cohort, a difference in the magnitude of PF-values was evident, whereas global as well as the general East-Asian population cohorts of ExAC and GnomAD were of low validity in the Japanese population (Appendix A).

### 2.3. Performance Indication of Consolidated BRCA1/2 Variant Interpretation

The overall measurement of MH BRCA performance in the test datasets was based on pooled *BRCA1/2* classifications (*n* = 1040) from four Japanese HBOC studies. Results showed 74.3% concordance with the interpretations of the examined studies and demonstrated particularly strong concordance within the pathogenic (98.1%) and benign (91%) subset (Figure 2I–K). Analysis of discordant interpretations showed that in the dataset of Momozawa et al. the missense variants *BRCA1* p.K1095E and *BRCA2* p.R174C (rs41293469), the small deletion *BRCA2* p.N1023_I1024del (rs730881605) and the last exon frameshift *BRCA2* p.N3407fs were reclassified in MH BRCA from pathogenic to VUS and the variant *BRCA2* p.R3052Q (rs80359171) to likely benign (Figure 2J and Appendix A). The latter variant was also classified as likely benign in the ARUP database relying on a genetic assessment study [22] and supported by curated evidence showing no damaging effect in the complementation assay, HR capacity, and cisplatin sensitivity [23]. Most contribution in discordance was due to VUS reclassification in 246 out of 582 variants with information gain for pathogenic (*n* = 2), likely pathogenic (*n* = 15), likely benign (*n* = 203), and benign (*n* = 26). Taken together, MH BRCA demonstrated its support in evidence-based *BRCA1/2* variant interpretation.

### 2.4. Comparison to ClinVar Submitter Content for BRCA1/2 Variant Classification

To evaluate the validity of the ACMG criteria interpretation made by the automated calculation in MH BRCA, we tested additional datasets of ClinVar submitter content of ENIGMA, ARUP, and the German Cancer Consortium (Figure 3A–C and Appendix A). Overall, in the three datasets, a concordance-rate of 99.98% was detected in the interpretation of pathogenicity with one discordant variant *BRCA1* p.L1365V (rs1567788936) in the German Cancer Consortium dataset (Figure 3C,D). A comparison of discordant variant classifications revealed an improvement for VUS reclassification in 32 out of 43 variants in the ARUP dataset (Figure 3B). In the ENIGMA dataset also 9 variants were reclassified with clinical impact by the tool (Figure 3A). Detailed analysis of the reasons for discordance demonstrated that variant annotations impacting the ACMG criteria PS3 and BS3 (well-established functional evidence) are important indicators for reclassification (Appendix A). However, the interpretation of likely benign variants varied massively, especially in the dataset of ENIGMA, for which possible reasons are discussed. In sum, the ACMG calculation made by MH BRCA confirmed its validity in the interpretation of pathogenicity.

#### 2.4.1. Clinical Evaluation of MH BRCA

To determine the clinical significance, we analyzed in MH BRCA retrospectively in 149 *BRCA1/2* variant positive cases from a total of 346 patients with breast, ovarian, or prostate cancer, which were sequenced at the YCH to test for an indication of HBOC or HPC syndrome, respectively. This included 142 breast and/or ovarian cancer and 7 prostate cancer cases with altogether 84 unique *BRCA1/2* variants. Clinical assessment at the YCH of *BRCA1/2* variant positive cases revealed deleterious *BRCA1/2* variants in 26 patients (17%) and variants classified as VUS in 34 patients (23%). A comparison of our results with classifications provided by MH BRCA demonstrated a complete overlap in the interpretation of pathogenicity and benignity (Figure 4A–C). The discrepancy in the interpretation of two likely benign variants were compensated by the information-gain for VUS reclassification in 17 out of 26 variants. Analysis of the clinical decision impact highlighted that one discordant case of clinical relevance would affect patient treatment decisions. Case analysis of the likely pathogenic reclassified VUS variant *BRCA1* p.V1653L (rs80357261) by MH BRCA revealed in our cohort three breast and/or ovarian cancer patients with otherwise no additional pathogenic event in the germline samples, which might be indicative for the disease-causing event (Table 1). Further analysis of variants reclassifying a VUS to likely benign were associated with 22 out of 34 patients (64.7%) with previously undetermined *BRCA1/2* variants for HBOC/HPC syndrome (Appendix A). Conclusively, MH BRCA indicated its clinical relevance due to improved reclassification of VUS.

#### 2.4.2. Correlation of Predicted Efficacy Association to PARP Inhibition by MH Guide

Finally, we examined to which degree MH BRCA classifications are representative for prediction to HR deficiency and response to PARP inhibition. To this end, we used the treatment decision support software MH Guide, providing treatment biomarkers and functional variant annotations. First, we assessed the outcome of our pre-evaluation test dataset from the examined four Japanese HBOC studies (Appendix A). Treatment efficacy association for a response to PARP inhibition was remarkable covered in 95.4% of classified pathogenic variants by MH BRCA. In contrast, 87.6% of classified benign variants correlated with the prediction of resistance to PARP inhibition due to functional evidence for a neutral variant (Appendix A). Although, VUS and likely benign classified variants by MH BRCA were largely not covered in MH Guide due to lack of functional evidence, yet 22 out of 357 VUS (6.2%) and 10 out of 296 likely benign (3.4%) variants were predicted to resistance to PARP inhibition (Appendix A). Interestingly, predicted low response to PARP inhibition due to hypomorphic variants conferring partially impaired moderate HR deficiency were reported in 11 variants classified as pathogenic, in 2 as VUS, 2 as likely benign, and 1 as benign.

Correlation of our clinical dataset of 149 *BRCA1/2* variant positive patient cases demonstrated complete coverage in the MH Guide of pathogenic and likely pathogenic variants with predicted response to PARP inhibition and benign variants with resistance to treatment (Figure 4D,E). In our dataset one variant, *BRCA2* p.I2675V (rs397507954), classified as likely pathogenic was predicted to be associated with low-efficacy to PARP inhibition [15]. Moreover, 1 out of 11 VUS and 1 out of 11 likely benign classified variants were predicted to confer resistance to treatment. Collectively, contextualizing the results of the consolidated *BRCA1/2* variant interpretation by MH BRCA with functional variant annotations of treatment biomarkers in the MH Guide highlighted an additional degree of information-gain for treatment response profiles.

## 3. Discussion

In this study, we sought to characterize the feasibility of using MH BRCA and MH Guide to support the clinical interpretation of NGS panel data from potentially HBOC-related cancer patients at the Yamanashi Central Hospital. Comparison to the test datasets of four examined HBOC studies and ClinVar submitter content of ENIGMA, ARUP, and German Cancer Consortium showed strong concordance in the assessment of pathogenicity. However, discordance in ACMG-guided interpretation of benign/likely benign classified variants in the ENIGMA dataset was evident. A potential reason is the ACMG criteria for the assessment of a synonymous (silent) variant for which splicing prediction algorithms predict no impact to the splice consensus sequence nor the creation of a new splice site and the nucleotide is not highly conserved (BP7). This criterion was not yet evaluated by MH BRCA during manuscript preparation. ENIGMA assessed the predicted splicing effects of synonymous variants by in silico calculation using the PRIORS tool of the University of UTAH. In addition, ENIGMA consortium decisions include also multifactorial likelihood models, which take into consideration family segregation data and trans/co-occurrence data, criteria which are not automatically assessed by MH BRCA due to case sensitive nature of such information. Discordance in benign classification to the study Momozawa et al. were mostly related to the ACMG criteria BS1, which incorporates information of allele frequency greater as expected for the disorder, and is automatically calculated by MH BRCA with a threshold of PF > 0.01% for HBOC-related genes in accordance to Kobayashi et al. [24].

In our clinical assessed *BRCA1/2* variant positive cohort the improvement of variant interpretation with MH BRCA was associated with detection of *BRCA1* p.V1653L (rs80357261), which showed a deleterious effect in cell viability assays [25], as a potentially pathogenic variant in three patients. This study of Findlay et al. assessed 3893 SNPs of *BRCA1* based on the saturation genomic editing method using CRISPR/Cas mediated high-throughput transformation of haploid HAP1 cells and measured the outcome in cell survival dependent on homologous recombination proficiency. Although the artificial nature of this high-throughput assay may be considered with caution (https://sge.gs.washington.edu/BRCA1/), a study investigating the impact of such proactive high-throughput functional assay indicated the usefulness for the reclassification of clinically observed VUS [26]. Moreover, another substitution at this residue in the BRCT domain, the *BRCA1* p.V1653M (rs80357261), showed a loss of transcriptional activity, slightly compromised protein binding abilities, and increased protease susceptibility, indicating a defect in protein folding [27]. For this reason, the percentage of patient cases with high HBOC risk increased from 26 to 29 patients (19% of cases), whereas none of the potentially HPC cases carried a deleterious *BRCA1/2* variant. The same incidence of HBOC syndrome was reported with 19.7% by the HBOC registration study of Arai et al. [14] but lower than found by Nakamura et al. with 30.7% in HBOC-associated breast cancer patients [28]. Refinement of patients with VUS into likely benign variants based on ACMG criteria strikingly reduced the uncertainty of germline-derived cancer-predisposing syndromes in 22 out of 34 patients. Recently, the corresponding variants were likewise predicted to be functionally neutral by a sequence-based computational model [29].

The potential degree of a predicted PARP inhibitor efficacy can be assessed in MH Guide based on experimental evidence for the impact on HR functionality due to hyper- or hypomorphic variants. In our patient cohort, the variant *BRCA2* p.I2675V (rs397507954) was reported to cause splicing defect with in-frame deletion resulting in a truncated protein of unknown impact on the vulnerability of HR activity [15]. In the validation dataset of the Japanese HBOC studies, an additional 11 out of 282 pathogenic classified variants (3.9%) by MH BRCA were predicted with low efficacy to PARP inhibition due to evidence for incomplete splicing defect or partial HR proficiency. However, conflicting experimental evidence or interpretation based solely on certain functional aspects, like compromised protein binding ability or reduced transactivation activity, complicate treatment decisions. Therefore, the predicted low efficacy to treatment considers variants to be monitored for potential inefficacy to PARP inhibition. Interestingly, two variants classified as benign in MH BRCA and Momozawa et al. [13], *BRCA2* p.T582P (rs80358457) and *BRCA2* p.R2108C (rs55794205), were predicted to have low response to PARP inhibition due to hypomorphic nature of the variants. Surprisingly, the variant *BRCA2* p.T582P (rs80358457) was classified in addition also by others as benign in multifactorial analysis or accordance with ACMG guidelines [30,31]. A review of the functional evidence provided by the MH Guide revealed a disturbing interaction with midbody proteins without affecting HR directed DNA repair activity [32]. However, the significance of the disruption of *BRCA2* interactions with midbody components remains to be established [33]. In contrast, a review of the variant *BRCA2* p.R2108C (rs55794205) elucidated an increased spontaneous intrachromosomal HR-directed DNA repair activity that may cause genomic instability [34]. It is plausible that variants associated with increased frequency of cytokinetic defects may contribute to pathogenicity, although not directly related to HR deficiency. Despite this discrepancy between classification and potential clinical impact, it must be emphasized that the benefit of the evidence-based approach of MH Guide and MH BRCA relies on the efficient support of clinicians in the interpretation of treatment relevant variant effects regarding the pathomechanism.

Given the challenge to keep track of the current state of published biomarker knowledge, especially in the light of the numerous high-throughput functional studies on *BRCA1/2* variants [25,35], such a curated evidence database based on published functional studies is of important clinical value [36]. Nevertheless, uncertainty in the variant interpretation of VUS, the discrepancy in the judgment of the pathogenic contribution, and lack of experimental evidence on variant impact or HR functionality remain key challenges for clinical interpretation. MH BRCA and MH Guide provide the first important step for an efficient, standardized, and transparent workflow in *BRCA1/2* variant interpretation.

## 4. Material and Methods

### 4.1. Patients and Methods

Peripheral blood samples were obtained from a total of 346 breast, ovarian and prostate cancer patients who attended Yamanashi Central Hospital (Yamanashi, Japan) between 2013 and 2018. Our cohort consisted of 334 breast and/or ovarian cancer patients with a high risk for HBOC. This cohort consisted of 239 patients with breast cancer, 15 with breast and ovarian cancer, and 80 with ovarian cancer. In addition, our study included 12 sporadic prostate cancer patients, which were assessed for potential *BRCA1/2* related hereditary prostate cancer (HPC) risk. Sample preparation is described elsewhere [15]. Informed consent was obtained from all subjects, and this study was approved by the institutional review board at Yamanashi Central Hospital on Jan 8th, 2012.

### 4.2. Targeted Next-Generation Sequencing (NGS)

For targeted NGS analysis, the Ion AmpliSeq^TM^
*BRCA1* and *BRCA2* Panel (Thermo Fisher Scientific, Waltham, MA, USA) containing 167 primer pairs in three pools or Oncomine *BRCA1/2* Panel (Thermo Fisher Scientific, Waltham, MA, USA) containing 275 primer pairs in two pools were used [37]. A detailed protocol of the procedure and analysis of sequenced data is described in our previous reports [15]. Massively parallel sequencing was carried out on a Personal Genome Machine (PGM) sequencer (Ion Torrent^TM^) using the Ion PGM Sequencing 200 Kit version 2 according to the manufacturer’s instructions. Sequence data were visually confirmed with the Integrative Genomics Viewer (IGV) and any sequence, alignment, or variant call error artifacts were discarded.

### 4.3. Data Analysis for Clinical Assessment of BRCA1/2 Interpretation

Classification of the deleterious variant was applied for frameshift, nonsense, and splice-site mutations that lead to premature truncation of the protein. Most of the cases were analyzed for their large deletion by the multiplex ligation-dependent probe amplification (MLPA) method which was carried out by FALCO Biosystems (Kyoto, Japan). Missense variants with minor allele frequency (MAF) <0.01 were selected as rare variants according to 1000 Genomes Project data [19], the 5000 Exome project (http://evs.gs.washington.edu/EVS/), and the Human Genetic Variation Database (HGVD) (http://www.genome.med.kyoto-u.ac.jp/SnpDB). Variants were annotated using the BIC database, the representative ClinVar database [17] and computational annotation systems (SIFT and PolyPhen2). Clinical features associated with HBOC have been determined based on personal and family history, histology of ovarian cancer (serous or non-serous) and breast cancer (triple-negative breast cancer [TNBC] or non-TNBC), age of developing breast cancer (≤45 years or older), and whether bilateral or unilateral breast cancer.

### 4.4. Study Design for Validation of MH BRCA

Comparison of *BRCA1/2* variant interpretations from our clinical assessed patient cohort and four Japanese HBOC studies, as well as the ClinVar submitter content from ENIGMA, ARUP laboratories, and the German Cancer Consortium was performed with MH BRCA (https://www.molecularhealth.com/eu/home/mh-brca/). ClinVar content was extracted by ClinVar Miner (https://clinvarminer.genetics.utah.edu/) version 2019-12. Results of MH BRCA classification from our patient cohort and the examined Japanese HBOC studies were tested for associated PARP inhibitor efficacy, using the treatment decision support service MH Guide (https://www.molecularhealth.com/eu/home/mh-guide/).

### 4.5. BRCA1/2 Interpretation by MH BRCA

MH BRCA technology combines ACMG guided interpretation with a proprietary database of expert-curated variant annotations and provides in addition to it a consensus classification of reputable variant annotation databases (ENIGMA, BIC, ClinVar, ARUP). Automated variant assessment on ACMG criteria is performed in accordance with the ruleset of the ACMG guideline [10] by calculation and interpretation of 17 out of 28 ACMG codes (Appendix A). To analyze *BRCA1/2* variants with the tool, a Variant Call Format (VCF) was generated, containing genomic coordinates of each variant based on the Reference Genome hg19/GRCh37 according to the nomenclature of the Human Genome Variation Society (HGVS). Translation of RefSeq transcripts of *BRCA1* (NM_007294.3) and *BRCA2* (NM_000059.3) into genomic HGVS coordinates were performed with ENSEMBL’s Variant Effect Predictor tool (http://grch37.ensembl.org/Tools/VEP). Classification accuracy was determined by cluster analysis of concordant and discordant interpretations in each classification subset (3- or 5-Tier system) and the concordance-rate was calculated. Discordant variant interpretations were depicted in a classification matrix. The clinical decision impact of discrepant classifications was defined by discordance in pathogenic and likely pathogenic variant interpretations.

### 4.6. BRCA1/2 Interpretation for Predicted Treatment Response to PARP Inhibition

MH Guide is a treatment decision support service to assist physicians in the clinical interpretation of NGS results coming from cancer patients and their tumors. The Dataome technology (https://www.molecularhealth.com/de/technologie/) provides an integrated analysis of a patient’s clinical and molecular cancer data derived from DNA sequence and searches the list of identified variants against a database of both established and emergent treatment biomarkers. These incorporate functional impact information of variants together with published treatment-related effects from regulatory agency recommendations, clinical trials, case reports, and pre-clinical studies. Following the Association of Molecular Pathology (AMP) guideline recommendations and by matching and contextualizing this knowledge with respect to the current patient condition, MH Guide provides clinicians with a prioritized list of drugs that are potentially effective, potentially ineffective, and/or potentially toxic for the patient. Prediction of the pathogenic effect of *BRCA1/2* variants and their potential DNA damage repair activity using curated functional evidence of the variant impact on protein, distinguishing hyper- and hypomorphic *BRCA1/2* variants.

### 4.7. Data Availability Statement

The authors declare that all relevant data supporting the findings of this study are available within the paper and its Appendix A.

### 4.8. Code Availability

The authors declare that the code or algorithm has restrictions to access due to CAP- and CLIA-certified commercial product registered as an IVD medical device in the EU.

## Figures and Tables

**Figure 1 ijms-21-03895-f001:**
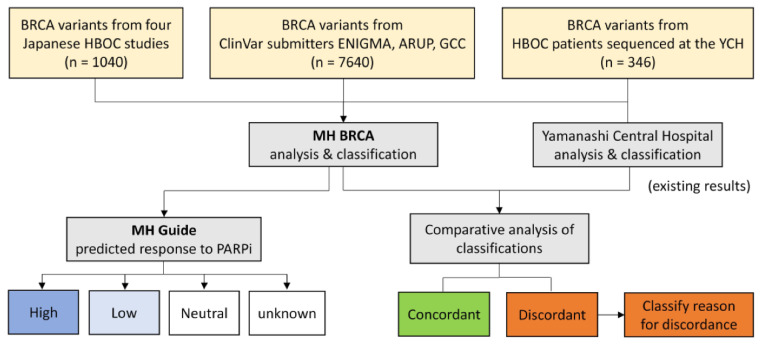
Study design for the validation of MH BRCA. *BRCA1/2* variants from Japanese HBOC studies of Arai et al. [12], Hirotsu et al. [13], and Nakagomi et al. [14] were compared to ACMG-guided classifications in MH BRCA with or without the Japanese genomic cohort data of ToMMo. The study of Momozawa et al. [15] was compared to a consolidated interpretation based on the database consensus classification and the ACMG-guided calculation provided by MH BRCA. Further, datasets of ClinVar submitters ENIGMA, ARUP, and the German Cancer Consortium were used to evaluate the ACMG classification of MH BRCA. Finally, we compared our clinically assessed *BRCA1/2* classifications from HBOC-related cancer patients sequenced at the Yamanashi Central Hospital. Comparative analysis in each dataset determined the concordance-rate and the reason of discordance. The datasets of our patient cohort and the examined Japanese HBOC studies were analyzed for predicted treatment response to PARP inhibition in MH Guide. Results were correlated to the corresponding BRCA1/2 variant classification by MH BRCA.

**Figure 2 ijms-21-03895-f002:**
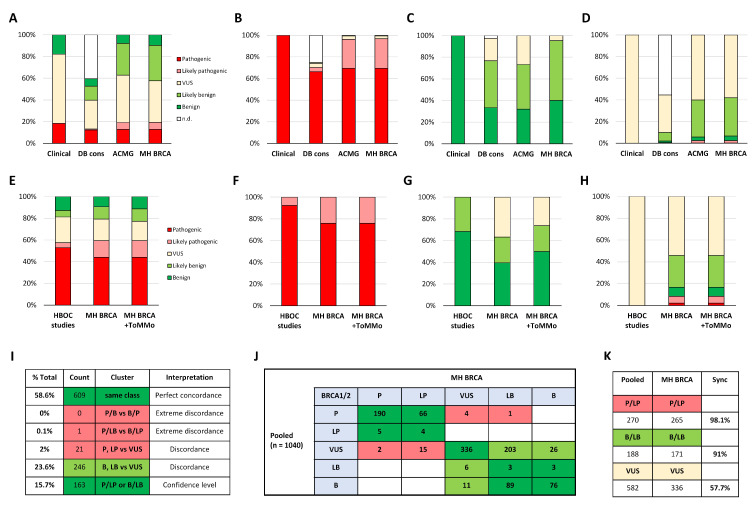
Impact of Japanese genomic cohort data evaluated in datasets of Japanese HBOC studies. (**A**) ACMG-guided classifications of 838 unique *BRCA1/2* variants collected from the study of Momozawa et al. [15] were compared to MH BRCA on the level of database consensus, ACMG-guided calculation and consolidated interpretation. Classification based on a 3 or 5-Tier system is shown in percentage. (**B**) Cluster analysis of 154 pathogenic classified variants. (**C**) Cluster analysis of 150 benign classified variants. (**D**) Cluster analysis of 534 VUS classified variants. (**E**) Classifications of 202 *BRCA1/2* variants were pooled together from three Japanese HBOC studies [12,13,14] and were compared to MH BRCA with or without the Japanese genomic cohort data of ToMMo. Classification based on the 5-Tier system is shown in percentage. (**F**) Cluster analysis of pooled pathogenic and likely pathogenic classified variants (*n* = 116), (**G**) cluster analysis of pooled benign and likely benign classified variants (*n* = 38), (**H**) cluster analysis of pooled VUS classified variants (*n* = 48). (**I**) Merged classifications of 1040 *BRCA1/2* variants from four Japanese HBOC studies. (**J**) Blotted classification-matrix illustrates concordance (green) and discordance affecting clinical decisions (red). (**K**) Percentage of concordance in each cluster. Abbreviations: P = pathogenic; LP = likely pathogenic; VUS = variant of unknown significance; LB = likely benign; B = benign.

**Figure 3 ijms-21-03895-f003:**
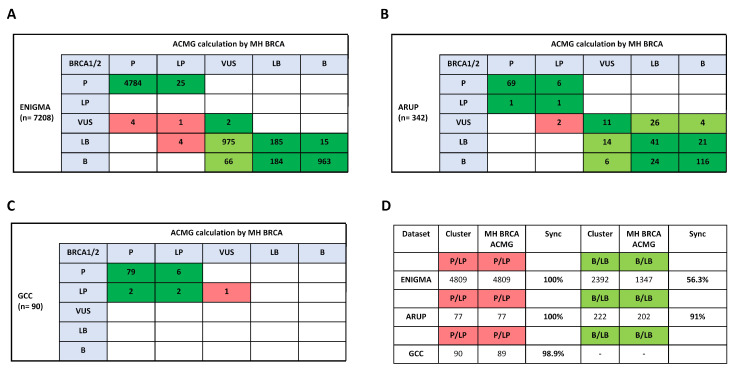
Comparison to ClinVar submitter content for *BRCA1/2* variant classification. Blotted classification-matrix illustrates concordance (green) and discordance of clinical significance (red). (**A**) ClinVar submitter content from ENIGMA included classifications of 7208 *BRCA1/2* variants. (**B**) ClinVar submitter content from the ARUP laboratories included classifications of 342 *BRCA1/2* variants. (**C**) ClinVar submitter content from the German Cancer Consortium included classifications of 90 *BRCA1/2* variants. (**D**) Concordance-rate in each dataset of examined ClinVar submitters is shown for the pathogenic/likely pathogenic and the benign/likely benign subset.

**Figure 4 ijms-21-03895-f004:**
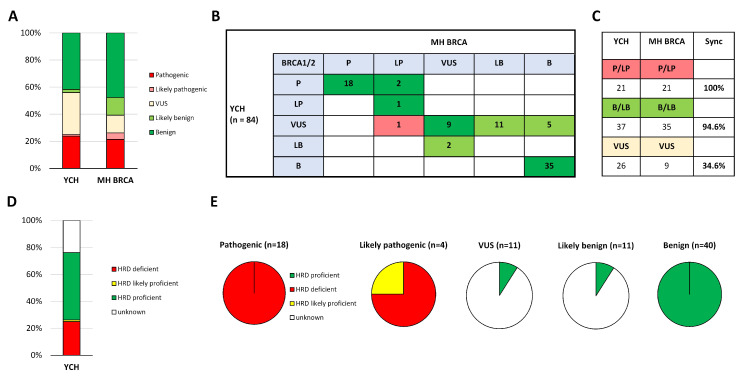
Clinical evaluation of MH BRCA. (**A**) Interpretations of 84 *BRCA1/2* variants clinically assessed at the Yamanashi Central Hospital (YCH) were compared with the results of MH BRCA classifications. 5-Tier system classification is shown in percentage. (**B**) Blotted classification-matrix illustrates concordance (green) and discordance of clinical significance (red). (**C**) Percentage of concordance in each cluster. (**D**) Predicted response to PARP inhibition in the MH Guide of 84 *BRCA1/2* variants is based on the functional impact of neutral, hypermorphic, or hypomorphic variants in homologous recombination activity and pathomechanism. (**E**) Cluster analysis of MH BRCA classified variants with a predicted response to PARP inhibition in MH Guide within the sub-categories. Abbreviations: P = pathogenic; LP = likely pathogenic; VUS = variant of unknown significance; LB = likely benign; B = benign.

**Table 1 ijms-21-03895-t001:** Case analysis of clinically relevant discordant *BRCA1/2* classification in MH BRCA. The *BRCA1* variant p.V1653L (rs80357261) was reclassified as likely pathogenic in MH BRCA and found in blood samples of 3 patients without any additional pathogenic germline *BRCA1/2* variant, which might be indicative for predisposition to HBOC syndrome.

Case-ID	Gene	HGVS g.	HGVS c.	HGVS p.	SNP-ID Number	YCH	MH BRCA
BRCA006	*BRCA1*	41222974C > A	4957G > T	V1653L	rs80357261	VUS	LP
	*BRCA2*	32914623G > T	6131G > T	G2044V	rs56191579	B	B
BRCA008	*BRCA1*	41222974C > A	4957G > T	V1653L	rs80357261	VUS	LP
	*BRCA2*	32914623G > T	6131G > T	G2044V	rs56191579	B	B
	*BRCA1*	41243841T > C	3707A > G	N1236S	rs863224760	VUS	VUS
	*BRCA2*	32910842A > G	2350A > G	M784V	rs11571653	B	B
BRCA052	*BRCA1*	41222974C > A	4957G > T	V1653L	rs80357261	VUS	LP

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
