# Peer review of "Consolidated BRCA1/2 Variant Interpretation by MH BRCA Correlates with Predicted PARP Inhibitor Efficacy Association by MH Guide"

_ijms, 2020, doi:10.3390/ijms21113895_

Round 1
Reviewer 1 Report
This paper deals with a very interesting issue: the BRCA1/2 variant interpretation to predict response to PARP inhibitors.The authors evaluate the performance of the so called MH BRCA and MH BRCA guide tools, developed by the Molecular Health company, on 346 Breast/ovarian /prostate cancer patients collected at the Yamanashi Central Hospital in Japan.
I found this paper interesting and well written. I suggest to: -do not use the word mutation but "pathogenic variant" -report the description of the critical ACMG criteria discussed instead of indicate only the ACMG code.
I have also few minor points to remark: - page 3 row 99 What is the FALCO Biosystems classification? - page 3 row 103 why the authors chose a Population Frequency>1%? --Page 5 paragraph 2.3.1 row180 deleterious variants are in 21 not 26 patients and VUS in
26 not 34 patients according to data reported in figure 4 B
Author Response
Response to Reviewer #1
Comments and Suggestions for Authors
This paper deals with a very interesting issue: the BRCA1/2 variant interpretation to predict response to PARP inhibitors.
The authors evaluate the performance of the so called MH BRCA and MH BRCA guide tools, developed by the Molecular Health company, on 346 Breast/ovarian /prostate cancer patients collected at the Yamanashi Central Hospital in Japan.
I found this paper interesting and well written. I suggest to: -do not use the word mutation but "pathogenic variant" -report the description of the critical ACMG criteria discussed instead of indicate only the ACMG code.
à We thank the reviewer for his advice for improvement and changed in accordance. We added in page 8 line 241-243 the sentence of the ACMG criteria BP7 as recommended.
I have also few minor points to remark: - page 3 line 99 What is the FALCO Biosystems classification?
à FALCO Biosystems is a Japanese test-laboratory for genetic analysis and has used until the end of 2019 the license of MYRIAD-Genetics (Salt Lake City, Utah, USA) for BRCA variant classification. The molecular diagnostic company MYRIAD uses their own proprietary database for interpretation. We specified in page 3 line 101 the above-mentioned information.
- page 3 line 103 why the authors chose a Population Frequency>1%?
à We use in context of the BRCA1/2 genes for BA1 a PF>1% in accordance with the ENIGMA classification criteria for class 1 (benign) variants (https://enigmaconsortium.org/wp-content/uploads/2018/10/ENIGMA_Rules_2017-06-29-v2.5.1.pdf). We integrated a new sentence in page3 line 104 to address this issue.
--Page 5 paragraph 2.3.1 line180 deleterious variants are in 21 not 26 patients and VUS in
26 not 34 patients according to data reported in figure 4 B
à The reported numbers in Fig.4B are related to unique BRCA1/2 variants found in the patient cohort. Thus, 26 pathogenic classified patients by the YCH have overall 21 unique pathogenic BRCA1/2 variants. From 34 patients with initially 26 VUS classified variants by the YCH have three patients a likely pathogenic variant (BRCA1.V1653L) classified by MH BRCA and the BRCA1/2 variants of the remaining 31 patients are shown in Supplemental Table 5.
Reviewer 2 Report
The study by Hirotsu et al. on MH BRCA1/2 variants classification and its impact on actionability is presented in a clear and interesting fashion.
The following changes are requested.
-While some rs accession numbers are given in Supplemental Table 4, it is recommended that all variants be identified with their rs numbers when available (in all Tables and Text) as this would improve variant identification, simplify tasks for the curious reader wanting more information on particular variants and enhance the visibility of the paper through entries in LitVar or other search engines. For example, providing the rs accession number for the variant BRCA2 R174C (rs41293469) allows a quick link to multiple entries in ClinVar and multiple citations in Pubmed that are not provided in the current manuscript.
https://www.ncbi.nlm.nih.gov/snp/?term=rs41293469
https://www.ncbi.nlm.nih.gov/clinvar/variation/51818/
-A main finding of this paper is the reclassification of the missense variant BRCA1-V1653L, detected in three patients, from a variant of uncertain significance (VUS) to a likely pathogenic (LP) variant with a PS3 evidence according to the ACMG guidelines. This reclassification results in a change in actionability with PARP inhibition becoming a chemotherapeutic option. This variant should be identified in the text with its full HGVS nomenclature. This finding should be reported in the abstract. The PS3 evidence comes from the proactive high-throughput Findlay functional study published in the journal Nature in 2018, which reported functional scores for 3,893 SNPs located within or near the 13 exons that encode the RING and BRCT domains of BRCA1 and defined 3 classes of variants (functional, intermediate, non-functional). In the current document, it is described that the reclassification comes from “deleterious in cell viability assays” (ref 26) (lane 250). Considering the importance of functional data in variant classification, it is desirable to critically describe the Findlay study and provide a link for more information/evaluation (https://sge.gs.washington.edu/BRCA1/). The recent publication by Kim et al. (PMID: 31907386) which also used the Findlay study to reclassify three BRCA1 variants (Asp96Glu (RING domain), Lys1711Glu and Leu1839Ser (BRCT domain)) should be included. The rs number for V1653L is rs80357261. The site is polymorphic and currently reported for the change V1653M (GRCh37.p13 chr17 NC_000017.10:g.41222974C>T NM_007294.3:c.4957 G>A). This change has been evaluated, for example, in a comprehensive analysis of missense mutations in the BRCT domain (PMID: 20516115). This information should be analyzed for accuracy in variant nomenclature/calling and possibly included if relevant.
-lane 31: changed 3,95 to 3.95
-lanes 90, 91: add commas for readability
-lanes 197-200: consider cutting sentence or adding commas for readability
Author Response
Response to Reviewer #2
Comments and Suggestions for Authors
The study by Hirotsu et al. on MH BRCA1/2 variants classification and its impact on actionability is presented in a clear and interesting fashion.
The following changes are requested.
-While some rs accession numbers are given in Supplemental Table 4, it is recommended that all variants be identified with their rs numbers when available (in all Tables and Text) as this would improve variant identification, simplify tasks for the curious reader wanting more information on particular variants and enhance the visibility of the paper through entries in LitVar or other search engines. For example, providing the rs accession number for the variant BRCA2 R174C (rs41293469) allows a quick link to multiple entries in ClinVar and multiple citations in Pubmed that are not provided in the current manuscript.
https://www.ncbi.nlm.nih.gov/snp/?term=rs41293469
https://www.ncbi.nlm.nih.gov/clinvar/variation/51818/
à We thank the reviewer for the recommendation to improve the data presentation. We have consistently added rs-numbers (SNP ID number) to all data tables and variants mentioned in the manuscript text.
-A main finding of this paper is the reclassification of the missense variant BRCA1-V1653L, detected in three patients, from a variant of uncertain significance (VUS) to a likely pathogenic (LP) variant with a PS3 evidence according to the ACMG guidelines. This reclassification results in a change in actionability with PARP inhibition becoming a chemotherapeutic option. This variant should be identified in the text with its full HGVS nomenclature.
à The variant is reported in the text with the rs-number and full HGVS nomenclature is provided in Table1 within the manuscript, as well as in the supplemental data table2 (YCH dataset).
This finding should be reported in the abstract.
à We integrated this information in the abstract in page 1 line 28 as: “…due to a likely pathogenic variant BRCA1 p.V1653L (NM_007294.3:c.4957G>T; rs80357261).”
The PS3 evidence comes from the proactive high-throughput Findlay functional study published in the journal Nature in 2018, which reported functional scores for 3,893 SNPs located within or near the 13 exons that encode the RING and BRCT domains of BRCA1 and defined 3 classes of variants (functional, intermediate, non-functional). In the current document, it is described that the reclassification comes from “deleterious in cell viability assays” (ref 26) (line 250). Considering the importance of functional data in variant classification, it is desirable to critically describe the Findlay study and provide a link for more information/evaluation (https://sge.gs.washington.edu/BRCA1/).
à We thank the reviewer for this suggestion. We added text (in page 8 line 256) with information regarding the SGE functional score experimental validation procedure.
The recent publication by Kim et al. (PMID: 31907386) which also used the Findlay study to reclassify three BRCA1 variants (Asp96Glu (RING domain), Lys1711Glu and Leu1839Ser (BRCT domain)) should be included.
à The study of Kim et al. is now cited at the newly added text in page 8 line 260.
The rs number for V1653L is rs80357261. The site is polymorphic and currently reported for the change V1653M (GRCh37.p13 chr17 NC_000017.10:g.41222974C>T NM_007294.3:c.4957 G>A). This change has been evaluated, for example, in a comprehensive analysis of missense mutations in the BRCT domain (PMID: 20516115). This information should be analyzed for accuracy in variant nomenclature/calling and possibly included if relevant.
à We would like to thank again the reviewer for his support to improve the manuscript and we are glad to add this reference. The information is added in page 8 line 262.
-line 31: changed 3,95 to 3.95
à done
-lines 90, 91: add commas for readability
à done
-lines 197-200: consider cutting sentence or adding commas for readability
à done
Reviewer 3 Report
The company Molecular Health (MH) developed MH BRCA, a technology that supports the clinical interpretation of BRCA1/2 variants by combining the rules laid out in the ACMG guidelines with a proprietary database of expert curated variant annotations and integrates variant population-frequency (PF) data from the Japanese genomic cohort of the Tohoku Medical Megabank Organization (ToMMo).
Major comments:
The authors remain very vague about the MH BRCA tool. It is a novel computational technology that combines the ACMG guidelines with expert curated variant annotations. It is therefore not very surprising that the overall concordance is high when you compare the output with ACMG based classifications (including many of the ClinVar classifications) and/or expert variant classifications.
Title does not match the content of the paper. This is about the concordance between MH BRCA based classifications and expert based variant classifications. The PARP inhibitor efficacy is predicted with another tool, the MH guide which is developed by the same company.
The currently developed classification guidelines, especially for BRCA1 and BRCA2 are mainly focused on germline variants and the association with cancer risk. At this moment, there is clearly a lack of information on the relationship with PARPi sensitivity. Although loss of function variants due to e.g. out of frame insertions/deletions or stopcodons confer sensitivity to PARPi there is at this moment little clinical data with regard to PARPi sensitivity for e.g. missense variants that affect but not completely abolish the function of the protein such as hypomorphic variants. The section 2.3.2. Correlation of efficacy association to PARP inhibition is therefore misleading. There has not been a study to correlate PARPi response in patients with MH BRCA based variant classifications. This is merely the correlation with the MH guide software. Also for this (commercially available) software which provides treatment biomarkers and functional variant annotation, it is unclear what the underlying algorithm is. In the abstract, the authors state “To assess whether classification results predict PARP inhibitor efficacy, contextualization with functional impact information on DNA repair activity were performed”. Although functional data are available in the public domain, none of the authors claim that their assays will predict PARPi sensitivity of tumors. They use the sensitivity of the variants to PARPi to address the impact on homologous recombination which is known to correlate with cancer risk.
Last sentence of discussion section “MH BRCA and MH Guide provide a first important step for an efficient, standardized and transparent workflow in clinically relevant biomarker detection”. What do the authors mean with this statement? Everybody responsible for variant classification uses the same (publicly available) information sources. Why focused on biomarker detection? In the abstract it is stated that MH BRCA is used for variant classification.
Minor points:
The authors should state explicitly that the tool does not take into account possible effect on RNA splicing. This is included in the discussion section but might be addressed also earlier.
Who is interpreting the functional data for PS3 and BS3 criteria? Use of MH Variant annotation database (VarDB) is indicated in supp Table1. Is this database publicly available?
The commonly used nomenclature according to Spurdle et al., (http:// dx. doi. org/ 10. 1136/jmedgenet- 2018- 105872) is pathogenic variants instead of mutations.
The discussion section line 288-294 about additional parameters such as mutational signatures is out of the scope of this manuscript.
Typo on the front page of your website https://www.molecularhealth.com/en/mh-brca/
“ focusing on germaline variants”
Author Response
Response to Reviewer #3
Comments and Suggestions for Authors
The company Molecular Health (MH) developed MH BRCA, a technology that supports the clinical interpretation of BRCA1/2 variants by combining the rules laid out in the ACMG guidelines with a proprietary database of expert curated variant annotations and integrates variant population-frequency (PF) data from the Japanese genomic cohort of the Tohoku Medical Megabank Organization (ToMMo).
Major comments:
The authors remain very vague about the MH BRCA tool. It is a novel computational technology that combines the ACMG guidelines with expert curated variant annotations. It is therefore not very surprising that the overall concordance is high when you compare the output with ACMG based classifications (including many of the ClinVar classifications) and/or expert variant classifications.
à We thank the reviewer for their constructive comments. Indeed, the high concordance serve as a validation of our tool. In specific, we mention that the MH BRCA tool incorporates as kind of “knowledge hub” the information from public variant annotation databases, summarized as database consensus classification (based on prioritization on ENIGMA = high, ClinVar 4 or 3 stars = high, ClinVar 2 stars and ARUP = mid, ClinVar 1 star or without = low), and in addition to it provides also an ACMG calculation. This automated classification incorporating different web-sources and algorithms for assessment of 17 out of 28 ACMG criteria (details provided in Supplemental Table 1). Interestingly, we demonstrated that this is already sufficient to classify variants of pathogenicity with high concordance to consortia evaluated variant classification by ENIGMA, as highlighted in figure 3A. We think that this is remarkable and not only as expected.
Title does not match the content of the paper. This is about the concordance between MH BRCA based classifications and expert based variant classifications. The PARP inhibitor efficacy is predicted with another tool, the MH guide which is developed by the same company.
à We specified in the title and the abstract the correlation to MH Guide.
The currently developed classification guidelines, especially for BRCA1 and BRCA2 are mainly focused on germline variants and the association with cancer risk. At this moment, there is clearly a lack of information on the relationship with PARPi sensitivity. Although loss of function variants due to e.g. out of frame insertions/deletions or stopcodons confer sensitivity to PARPi there is at this moment little clinical data with regard to PARPi sensitivity for e.g. missense variants that affect but not completely abolish the function of the protein such as hypomorphic variants. The section 2.3.2. Correlation of efficacy association to PARP inhibition is therefore misleading.
à We thank again for this critical view regarding the variant classification guidelines and PARPi sensitivity. MH BRCA as tool for classification of germline variants from blood-samples classify variants based on the ACMG guidelines. In contrast, the MH Guide logic follows the AMP guidelines (PMID: 27993330), which are based on clinical actionability and pathobiological relevance. We would like to argue that regulatory agencies, like the FDA, EMA, or PMDA, have approved deleterious or suspected deleterious BRCA1/2 variants for several PARP inhibitors as olaparib, rucaparib, and talazoparib in various indications. These regulatory approvals are based on several clinical studies and we therefore not understand the “clearly lack of information on the relationship with PARPi sensitivity”. However, we agree with the reviewer comment on hypomorphic variants and see a need for better information of clinicians regarding the functional impact of missense variants. Molecular Health provides by MH Guide a diversification of BRCA1/2 variants into hyper- or hypomorphic variants based on functional impact information, a first and important basis to investigate the efficacy association to PARPi in potentially future clinical studies. We changed the title of the corresponding section (2.3.2) to predicted correlation by MH Guide.
There has not been a study to correlate PARPi response in patients with MH BRCA based variant classifications. This is merely the correlation with the MH guide software. Also for this (commercially available) software which provides treatment biomarkers and functional variant annotation, it is unclear what the underlying algorithm is.
à The MH Guide algorithm is based on the Dataome technology (weblink see methods and material section), which provides an integrated analysis of a patient’s clinical and molecular cancer data derived from DNA sequence and searches the list of identified variants against a database of both established and emergent treatment biomarkers. Such clinical variant annotations (CVIs) incorporate functional impact information of variants together with published treatment related effects from regulatory agency recommendations, clinical trials, case-reports, and pre-clinical studies. We added in page 10 line 367 more detailed information in the methods and material section.
In the abstract, the authors state “To assess whether classification results predict PARP inhibitor efficacy, contextualization with functional impact information on DNA repair activity were performed”. Although functional data are available in the public domain, none of the authors claim that their assays will predict PARPi sensitivity of tumors. They use the sensitivity of the variants to PARPi to address the impact on homologous recombination which is known to correlate with cancer risk.
à We would like to argue that only the minority of functional data in the public domain are related directly to measure the sensitivity to PARPi (PMID: 29988080 see Figure S9). Most of the publications, which investigate BRCA1/2 functional impact, assess homologous recombination activity for example by the RECAP assay (PMID: 30139880), Rad51 foci formation (PMID: 31569370), GFP reporter plasmid assay (PMID: 29884841), or assess other impact important for the functionality of BRCA1/2 genes like protein binding abilities by the GFP-reassembly assay (PMID:30696104), luciferase gene reporter assay (PMID: 29236234), or splicing effects (PMID: 30883759). Since such well-established functional studies are used for ACMG criteria PS3 and BS3, it also provides the information for deleterious or suspected deleterious variants as recommended by the agencies for PARPi approval. We added in the abstract the specific use of MH Guide in page 1 line 30.
Last sentence of discussion section “MH BRCA and MH Guide provide a first important step for an efficient, standardized and transparent workflow in clinically relevant biomarker detection”. What do the authors mean with this statement? Everybody responsible for variant classification uses the same (publicly available) information sources. Why focused on biomarker detection? In the abstract it is stated that MH BRCA is used for variant classification.
à We agree with the reviewer’s impression and changed the sentence in accordance with the suggested recommendation in page 9 line 304.
Minor points:
The authors should state explicitly that the tool does not take into account possible effect on RNA splicing. This is included in the discussion section but might be addressed also earlier.
à This information is now added in page 8 line 243 (BP7 criteria). However, splicing effects are supported by MH BRCA due to the criteria PS3 and BS3 reporting functional assay on splicing activity.
Who is interpreting the functional data for PS3 and BS3 criteria? Use of MH Variant annotation database (VarDB) is indicated in supp Table1. Is this database publicly available?
à Functional impact and treatment response information from published peer-reviewed evidences is curated at Molecular Health by a team of experts with doctoral-degree in medical and natural science. Our review process is based on at least two scientific curators (author and QA-reviewer) and a final medical review by an experienced physician. VarDB is a proprietary in-house database of Molecular Health for curation of protein impact information.
The commonly used nomenclature according to Spurdle et al., (http:// dx. doi. org/ 10. 1136/jmedgenet- 2018- 105872) is pathogenic variants instead of mutations.
à In accordance with reviewer 1, too, we have consistently changed the term mutation to variant.
The discussion section line 288-294 about additional parameters such as mutational signatures is out of the scope of this manuscript.
à We agree with the reviewer recommendation and deleted these sentences.
Typo on the front page of your website https://www.molecularhealth.com/en/mh-brca/
“ focusing on germaline variants”
à We appreciate this feedback and have addressed the responsible department for implementing the change. Thank you!
Round 2
Reviewer 3 Report
I'm satisfied with the response of the authors but I sincerely regret that the commercial use of the tool for automated variant classification will restrict the use for a broad genetic society.